# *Prunella vulgaris* Extract Ameliorates Testosterone-Induced Benign Prostatic Hyperplasia by Regulating Androgen Levels, Cell Proliferation, and Apoptosis

**DOI:** 10.3390/ph17111516

**Published:** 2024-11-11

**Authors:** Poornima Kumbukgahadeniya, Eun-Bok Baek, Eun-Ju Hong, Jun-Yeop Song, Youn-Gil Kwak, Mi-Ran Jang, Hyo-Seong Ji, Hyo-Jung Kwun

**Affiliations:** 1Department of Veterinary Pathology, College of Veterinary Medicine, Chungnam National University, Daejeon 34134, Republic of Korea; poornima.lakshini93@gmail.com (P.K.); baekeunbok@hanmail.net (E.-B.B.); bioeun59@naver.com (E.-J.H.); qldqld0610@gmail.com (J.-Y.S.); 2Huons Foodience Co., Ltd., 19, Insam Gwangjang ro, Geumsan eup, Geumsan gun 32724, Republic of Korea; kyg@huonsfoodience.com (Y.-G.K.); rose86mr@huonsfoodience.com (M.-R.J.); hsji@huonsfoodience.com (H.-S.J.)

**Keywords:** benign prostatic hyperplasia, *Prunella vulgaris* L., proliferation, apoptosis, growth factors

## Abstract

Background/Objectives: Benign prostatic hyperplasia (BPH) is a prevalent urological condition affecting elderly men. *Prunella vulgaris* L. (PV), a perennial herbaceous plant native to Europe and Asia, has anti-inflammatory, antioxidant, and antimicrobial effects. In this study, we determined the effect of PV extract on the development of BPH. Methods: Rats were treated via a daily hypodermic injection of testosterone propionate (TP; 3 mg/kg) for 4 weeks. Groups of BPH rats were treated with or without PV (60 or 80 mg/kg) by oral gavage. Results: In BPH model rats, PV considerably reduced their relative prostate weight and serum concentrations of dihydrotestosterone (DHT) and testosterone. The TP-induced increases in epithelial thickness in the prostate, proliferating cell nuclear antigen (PCNA) expression, and cyclin D1 expression were remarkably reduced, whereas terminal deoxynucleotidyl transferase dUTP nick-end labeling (TUNEL)-positive cells and cleaved caspase-3 levels were increased, in PV-treated rats compared to BPH rats. The mRNA expression levels of growth factors, such as transforming growth factor-β (TGF-β), fibroblast growth factor (FGF), and insulin-like growth factor (IGF-2), were significantly reduced in PV-treated rats. Mechanistically, the TP-induced activation of c-Jun N-terminal kinase (JNK) was reduced by PV administration. Conclusions: These results designate that PV effectively ameliorates the development of testosterone-induced BPH through anti-androgenic, anti-proliferative, and pro-apoptotic activities, suggesting that it could be a potential therapeutic substance for BPH.

## 1. Introduction

Benign prostatic hyperplasia (BPH) is a prevalent urological condition affecting elderly men, with an estimated prevalence of 8% at 40, 50% at 60, and 80% at 90 years of age [1]. BPH involves non-malignant, hyperplastic, and progressive histopathological changes in the prostate’s transitional zone; this causes prostatic nodule formation, inflammation, fibrosis, and alterations in smooth muscle function, leading to urethral blockage and lower urinary tract symptoms (LUTS) [2,3]. The exact molecular mechanism underlying BPH remains uncertain, but the increased cell count may be due to impaired programmed cell death and epithelial and stromal cell proliferation [4]. The normal development and functions of the prostate depend on the reduction of testosterone to dihydrotestosterone (DHT), which interacts with androgen receptors (ARs) to promote protein synthesis and prostate cell proliferation [5,6,7]. As males age, testosterone levels decline; BPH patients, however, experience increased DHT secretion and a consequent enlargement of prostate volume [8,9]. Several studies have identified growth factors involved in BPH, including transforming growth factor beta (TGF-β) family members, fibroblast growth factor (FGF), and insulin-like growth factor type (IGF) [10,11]. These growth factors regulate the signaling among epithelial cells and stroma in the prostate gland to regulate cellular proliferation and apoptosis [12].

Treatment options for symptomatic BPH include lifestyle changes and/or medical, device-based, or surgical therapies [13,14,15]. The drugs utilized, which may be used alone or in combination, include 5-alpha-reductase antagonists (5-ARIs), alpha-adrenergic antagonists (α1-blockers), and phosphodiesterase-5 enzyme antagonists. The most used medications are 5-ARIs, α1-blockers, and cocktails of the two combined (tamsulosin and dutasteride) [16,17,18]. However, these medications have potential adverse effects such as a decreased libido, erectile dysfunction, gynecomastia, depression and anxiety, and an increased risk of high-grade prostate cancer, etc. [19]. Due to these side effects, the therapeutic application of plant extracts, called phytotherapy, has become an alternative approach for managing BPH [20]. Phyto therapeutic agents are common in Europe, and their use is increasing in Western populations [20].

*Prunella vulgaris* L. (PV) is a perennial herbaceous plant that is native to Europe and Asia and grows in temperate regions worldwide, including Korea [21]. PV has long been used as a therapeutic agent in conventional medicine, such as in treating thyroid dysfunction, pulmonary tuberculosis, mastitis, arterial hypertension, and infectious hepatitis [22,23]. PV can also be used in patients with goiter, dermatitis, and dermal allergy. It shows anti-allergic, antioxidant, antimicrobial, and anti-inflammatory activities [23,24]. While PV extract is generally considered safe, it is important to be aware of potential side effects such as gastrointestinal discomfort and allergic reactions [25]. PV’s main components include phenolic acids, mono- and sesquiterpenoids, higher fatty acids, pentacyclic triterpenes, flavonoids, vitamins, polysaccharides, nitrogen-containing compounds, and tannins. The bioactive components of PV include caffeic acid, rosmarinic acids, oleanolic acid (OA), and ursolic acid (UA) [21,23,24]. Rosmarinic acid, a combination ester of lactic and caffeic acid, is an effective natural antioxidant against free radical pathologies, such as atherosclerosis, radiation sickness, and cancer [23,24]. It binds oxygen free radicals and inhibits UVA-modulated protein expression [24]. OA and UA are classified as pentacyclic triterpenoids and have been extensively studied in vitro and in vivo for their biological activities [26].

While there is no specific research directly linking PV extract to BPH treatment, its potential anti-inflammatory, antiproliferative, and antioxidant properties could be beneficial. These properties may help to reduce prostate inflammation and oxidative stress, which are often associated with BPH [27]. Previous studies highlighted that PV extract can induce cancer cell apoptosis, inhibit angiogenesis, and suppress tumor cell migration and invasion, and PV-containing herbal mixtures can remodel the tumor immune microenvironment by increasing anti-tumor immune cells and suppressing immune-suppressive cells [25,28]. This can enhance the effectiveness of cancer immunotherapy. Based on this literature, it can be concluded that PV is a potential candidate in treatments for BPH. Here, we utilized a testosterone-induced rat BPH model to determine the medicinal effects of PV against BPH and their fundamental molecular mechanism(s).

## 2. Results

### 2.1. HPLC Analysis of PV Extract

The triterpenoids in PV extract were identified using HPLC analysis. OA and UA were determined by comparing the retention times of peaks in the crude extract to those of known standards and were detected in significant amounts (2.03 ± 0.01 and 7.03 ± 0.03 mg/g of extract) (Figure 1). The results showed that OA and UA are major compounds of PV extract.

### 2.2. PV Extract Reduces Prostate Relative Weight in Rat BPH Model

Prostate relative weights were increased in the BPH group (0.346 ± 0.075%) compared to the NC group (0.185 ± 0.031%). Compared to the BPH group, significantly lower prostate weights were seen in groups treated with finasteride (0.292 ± 0.059%), PV at 60 mg/kg (0.289 ± 0.064%), and PV at 80 mg/kg (0.287 ± 0.053%) (Table 1). The percent inhibition of prostate gland growth in the finasteride (BPH + FIN) and PV (BPH + PV60, BPH + PV80) groups were 66.49%, 35.87%, and 36.64%, respectively (Table 1). There was no statistically significant variation in the body mass of the rats across the different experimental groups.

### 2.3. PV Extract Downregulates the Serum Levels of DHT and Testosterone in a Rat BPH Model

The BPH group exhibited significant increases in serum levels of DHT and testosterone in comparison to the NC cohort, and those levels were significantly lower in the finasteride- and PV-treated cohort than in the BPH cohort (Figure 2).

### 2.4. PV Extract Reduces the Epithelium Thickness of the Prostate Gland in a Rat BPH Model

The prostate glands of the NC group showed no histological change, whereas the BPH group exhibited increases in epithelial hyperplasia and glandular epithelial thickness (Figure 3). Treatment with finasteride and PV extract significantly reduced the thickness of the prostatic epithelium in comparison to that seen in the BPH group (Figure 3).

### 2.5. PV Extract Suppresses the Proliferation of Prostate Cells in a Rat BPH Model

Immunohistochemical staining showed that the count of proliferating cell nuclear antigen (PCNA)-positive cells was significantly increased in the BPH cohort compared to the NC cohort (Figure 4A,B), whereas the numbers in the finasteride and PV cohorts were considerably less than that in the BPH group (Figure 4A,B). Western blot analysis showed that cyclin D1 expression was significantly upregulated in the BPH group compared to NC, but this change was alleviated in the groups of finasteride and PV (Figure 4C).

### 2.6. PV Extract Elevates Apoptosis of Prostatic Cells in Rat BPH Model

Compared with the NC batch, the count of terminal deoxynucleotidyl transferase dUTP nick-end labeling (TUNEL)-positive cells was reduced in the BPH cohort. However, the TUNEL-positive cell count was remarkably enhanced in the finasteride and PV groups relative to the BPH group (Figure 5A,B). Western blot analysis showed that the expression level of cleaved caspase-3 was upregulated in the PV groups compared to the BPH group (Figure 5C).

### 2.7. PV Extract Decreases the Expression Levels of Growth Factors in a Rat BPH Model

The mRNA expression levels of IGF-2, TGF-β, and FGF in the BPH cohort were increased compared to those in the NC cohort; nevertheless, the finasteride and PV groups showed significant reductions in the expression levels of IGF-2, TGF-β, and FGF compared to those in the BPH cohort (Figure 6).

### 2.8. PV Extract Reduces JNK Activation in a Rat BPH Model

The phosphorylation of c-Jun N-terminal kinase (JNK) (indicating its activation) was elevated in the BPH group compared with the NC group but reduced in the finasteride and PV groups compared to the BPH group (Figure 7).

## 3. Discussion

BPH is a degenerative disorder characterized by prostatic hypertrophy with LUTS [2]. Our current study focused on whether PV extract exhibits protective effects in testosterone-induced BPH model rats. Our analytic results revealed that PV extract attenuates the testosterone-mediated elevations in relative prostate weight, serum testosterone, and serum DHT in BPH model rats. The administration of PV extract also markedly reduced testosterone-induced increases in prostatic glandular epithelial cells and growth factor expression and activated apoptotic signaling pathways in BPH model rats. These findings suggest that PV extract inhibits the advancement of testosterone-induced BPH.

Testicular androgens, such as testosterone and DHT, are essential for prostate development [29,30]. DHT is predominantly generated by 5α-reductase (5αR) from testosterone; it binds androgen receptors with high affinity, and the DHT–androgen receptor complex initiates the transcription and ultimately regulates the cell cycle, cell growth, and differentiation of prostate stromal and epithelial cells to promote enlargement of the prostate and BPH pathogenesis [29]. Finasteride is a 4-azasteroid antagonist of 5αR [31] that functions by forming a stable complex with, and thereby selectively inhibiting, the enzyme 5αR [30] to obstruct the peripheral transformation of testosterone into DHT. This substantially reduces the tissue and serum concentrations of DHT, leading to a concomitant decrease in prostatic size [32]. Despite these beneficial actions, however, finasteride is associated with significant adverse effects. This has prompted researchers to explore alternative therapeutic agents that may be utilized to manage BPH while minimizing side effects [33]. Here, we show that BPH model rats exhibit increases in serum levels of testosterone and DHT, but this is ameliorated by treatment with PV extract.

One of the most well-known causes of BPH is an imbalance between cell proliferation and death [34,35]. In the cellular proliferation process, PCNA critically contributes to DNA replication and repair, while nuclear cyclin D1 regulates cell cycle progression [36,37]. Our present results revealed that PV treatment attenuated the relative protein expression level of cyclin D1 in the BPH model and significantly decreased the numbers of PCNA-positive nuclei on immunohistochemical analysis, in comparison to these parameters in the BPH group. Regarding apoptosis, we examined the level of cleaved caspase 3, which is an important tumor gene and apoptotic protein [10,12], and found that cleaved caspase 3 was downregulated in the BPH cohort, and this outcome was ameliorated in the finasteride- and PV-treated categories. This suggests that both treatments are inducing apoptosis in prostate cells. The fact that the NC and BPH groups show similar levels of cleaved caspase-3 indicates that BPH itself may not significantly alter the apoptotic rate in prostate cells. However, both finasteride and PV extract seem to enhance apoptosis. According to the antiproliferative results, finasteride and PV extract primarily induce apoptosis, while significantly affecting cell proliferation; this could be beneficial in reducing prostate enlargement. However, with regards to the apoptotic effect of PV, previous studies demonstrated that PV extract induced apoptosis by upregulating the expression of cleaved caspase-3, along with other apoptotic markers like Bax [38]. While the results presented in Figure 5C suggest that PV extract can induce apoptosis in prostate cells, further investigation is needed to determine the optimal balance between cell proliferation and apoptosis to achieve effective BPH treatment. The numbers of TUNEL-positive cells were also increased in the PV- and finasteride-administrated categories, relative to the BPH cohort. We herein identified UA as a major compound of PV extract. A previous study showed that UA is nontoxic against healthy cells but exhibits apoptotic effects against cervical, breast, and colorectal cancer cells [26]. Here, we provide evidence that PV extract attenuates cell proliferation and increases apoptosis in testosterone-induced BPH model rats.

Growth factors, particularly members of the FGF, IGF, and TGF-β families, are crucial for the development and progression of BPH [39,40]. FGF has been identified in BPH stromal cells and prostate cancer epithelial cells, where it plays significant roles in tumor growth, cellular proliferation, and prostatic growth [41]. Previous studies examining the relevant roles of growth factors (e.g., IGFs) have suggested that they may contribute to the development of BPH through androgen-independent mechanisms [42,43]. High IGF-1 blood levels have been associated with prostatic malignancy [44], while IGF-2 reportedly promotes cellular proliferation and growth in the prostate gland [45]. T lymphocyte-secreted TGF-β plays important roles in regulating the proliferation and differentiation of the prostate stroma [39,46]. Previous studies showed that sulfated polysaccharide extracted from PV attenuates the expression of FGF on endothelial cells [47] and that PV can downregulate TGF-β expression in models of renal injury and extrinsic aging [48]. Here, we report that the levels of FGF, IGF-2, and TGF-β are significantly higher in the prostates of BPH rats compared to those of the NC group, and these differences are notably rescued in the PV group compared to the BPH group. Our present findings suggest that PV may potentially alleviate BPH by influencing growth factor responses to regulate cellular proliferation and prostatic growth.

The MAPK family member, JNK, is activated by phosphorylation and acts as a vital serine/threonine protein kinase for normal cell proliferation and development [49]. Some studies have suggested that JNK is regulated in the enlarged prostate and/or during the proliferation of prostate stromal cells [50], while others found that an interaction with AR triggers JNK phosphorylation and downstream signaling to facilitate cell cycle progression in BPH and prostate cancer [51]. Testosterone and DHT are reported to rapidly and transiently activate MAPK in cultured hippocampal neurons, as evidenced by the phosphorylation of JNK [52]. The JNK signaling pathway is critical for FGF-induced epithelial cell proliferation [53] and IGF and TGF-β induced cellular proliferation [54], all of which are important for the development of BPH. Previous reports found that PV extract can attenuate phospho-JNK levels in various cell types, including INS-1 cells and normal human dermal fibroblasts [55]. Here, we show that the phosphorylation of JNK is increased in the prostate tissues of BPH rats, and this change is significantly ameliorated by PV extract.

In our study, we tested multiple concentrations of PV extract against BPH rats. The highest concentration, 80 mg/kg, achieved effective results in many parts of the study. While increasing the dosage of PV above 80 mg/kg might seem like a straightforward approach to enhance its effectiveness, it is important to consider the potential risks and limitations associated with higher doses, such as increased toxicity, the saturation of receptors, a non-linear dose–response relationship, pharmacokinetic difference, and pharmacodynamic difference [38]. Further studies are needed to determine an accurate dosage of PV extract as a therapeutic agent for BPH.

The most used medications are 5-ARIs, α1-blockers, and cocktails of the two combined (tamsulosin and dutasteride), and the most common mechanism of these drugs is inhibiting the conversion of testosterone to DHT by inhibiting the 5αR enzyme [16,17,18]. According to the previous literature, for treating BPH, the most used herbal plants are *Serenoa repens, Prunus africana*, *Urtica dioica*, *Cucurbita pepo*, and *Secale cereale* [56,57]. Based on earlier studies, these plants have similar mechanisms of action, such as inhibiting the conversion of testosterone to DHT, and anti-inflammatory effects [56,57]. According to our study, PV extract shows potential in treating BPH. While it is not necessarily “more effective” than all other plant materials, it possesses several properties that make it a promising option, such as regulating androgen levels, antiproliferative effects, apoptosis, and anti-growth factor effects. It can be a potential candidate in treating BPH, as a multi-targeted natural product.

## 4. Materials and Methods

### 4.1. Preparation of PV Extract

PV was purchased from Jecheon Herbal Market (Jecheon, Chungbuk, Republic of Korea), and voucher specimens (No. 2022-HF001, confirmed) were deposited at the college of pharmacy, Han Yang University (Seoul, Republic of Korea). PV was extracted as follows: dried PV spikes were macerated in 70% ethanol at 50 °C for 8 h, the remnant was extracted a second time with 50% ethanol at 50 °C for 15 h, the two infusions were mixed and concentrated under reduced pressure, and the concentrate was lyophilized. The PV extract was produced in Huons Foodience under principles of Good Manufacturing Practice (GMP) and stored at −20 °C for further analysis.

### 4.2. HPLC Analysis

The extract (1 g) was dissolved in a 100-mL volumetric flask using methanol, filtered through a 0.45 μm PDVF filter, and injected into an Agilent 1260 series HPLC system equipped with a DAD (diode array detector). Chromatographic separation was performed on an Acclaim C30 column (250 × 4.6 mm, 5 μm) with the column temperature maintained at 25 °C. The detection wavelength was set at 210 nm, and the injection volume was 10 μL. An isocratic mobile phase of methanol/0.2% ammonium acetate was used, with a flow rate of 0.4 mL/min. The standards of OA (O5504) and UA (U6753), (purity 99% or higher) were purchased from Sigma Aldrich (Saint Louis, MO, USA). The OA and UA in the extract were identified based on their retention times and spectrums, compared with those of their respective standards.

### 4.3. Animals

Sprague Dawley male rats *(n* = 25, 7 weeks old) were obtained from Orient Bio (Seongnam, Republic of Korea), and the mice were divided into numbers of 5 for each group and maintained under standard laboratory conditions (22 ± 2 °C; relative humidity, 50 ± 5%; 12 h light/dark cycle); standard rodent dietary formulations were fed to them, and they had ad libitum access to sterilized tap water. The methodologies for the experimental animal study received endorsement from the Animal Experimentation Ethics Committee at Chungnam National University in Daejeon, Republic of Korea.

### 4.4. Initiation of BPH in Rats and Treatment Design

Rats were randomly divided into four groups and administrated the following dosage regimens: (1) the NC group received a subcutaneous (SC) injection of corn oil (as a vehicle control) and oral gavage of PBS (as a vehicle control); (2) the BPH group received an SC injection of 3 mg/kg TP (Tokyo Chemicals Ind. Co., Chuo ku, Tokyo, Japan) and oral gavage of PBS; (3) the BPH plus finasteride (BPH + FIN) group (positive control), received an SC injection of TP and oral gavage of finasteride (10 mg/kg; Sigma); and (4) the BPH + PV group received an SC injection of TP and oral gavage of PV (60 mg/kg or 80 mg/kg). All treatments were given daily for 4 weeks; the treatment volumes were 5 mL/kg for oral gavage (PBS, finasteride, and PV) and 3 mL/kg for SC injections. The lyophilized PV extract was dissolved in PBS. A single batch of PV was prepared for the rats’ oral gavage, which was adequate for a 4-week duration. The body mass of the experimental rats was assessed on a weekly basis throughout the duration of the study, in addition to being recorded on the day designated for necropsy. Following the final administration of the injection, the rats were humanely euthanized, and blood samples were procured from the vena cava. Serum was subsequently extracted from the whole blood and preserved at a temperature of −70 °C. The prostate tissues were meticulously excised and weighed.

### 4.5. Analysis Serum of DHT and Testosterone Levels

Commercial assay kits were used to determine the serum levels of testosterone (Cayman Chemicals, Ann Arbor, MI, USA) and serum DHT (ALPCO Diagnostics, Salem, NH, USA). All procedures were performed according to the manual. In summary, the serum specimens underwent incubation with enzyme conjugates, were washed, and exposed to substrates, and the testosterone and DHT concentrations were determined by assessing absorbance at 412 and 450 nm, respectively.

### 4.6. Histopathology

Prostatic tissues were preserved in a 10% buffered formalin solution, subsequently embedded in paraffin wax, sectioned to a thickness of 4 μm, and subjected to staining with hematoxylin and eosin (H&E). The measurement of epithelial thickness was conducted in the ventral lobe of each prostate in accordance with previously established methodologies [13]. To summarize, the epithelial structures in five randomly selected microscopic fields (×400) from each rat (*n* = 5 per group) were quantified utilizing Image J software (version 46a; NIH, Bethesda, MD, USA).

### 4.7. Immunohistochemical (IHC) Staining of Proliferative Cell Nuclear Antigen (PCNA)

Briefly, sections were de-paraffinized by xylene and rehydrated in an ethanol series. Following the process of heat-induced antigen retrieval, the specimens underwent blocking with normal goat serum (Vectastain, Cambridge, UK) and were subsequently incubated overnight in the presence of a primary antibody targeting proliferative cell nuclear antigen (PCNA, dilution 1:10,000; Abcam, Cambridge, UK), alongside a secondary antibody (Vectastain). The detection of avidin–biotin complexes was accomplished utilizing diaminobenzidine (DAB; Vector Laboratories, Newark, CA, USA) reagent. Five randomly selected regions (×400) were observed for each rat (*n* = 5 per group), and the numbers of PCNA-positive cells per 100 cells were counted.

### 4.8. TUNEL Assay

The TUNEL assay was performed using a commercial kit (Merck Millipore Corporation, Burlington, MA, USA) according to the manufacturer’s manual. Briefly, sections were de-paraffinized by xylene, rehydrated with an ethanol series, pretreated with Proteinase K enzyme (Merck Millipore Corporation, MA, USA), and subjected to the inactivation of endogenous peroxides using 3.0% hydrogen peroxide (SAMCHUN Chemical Co., Ltd., Bongeunsa ro 37 gil, Seoul, Republic of Korea). Biotinylated dUTP was incorporated onto the 3′ ends of fragmented DNA in a reaction mixture containing TdT enzyme (Merck Millipore Corporation), DAB reagent was used to observe the immune complexes, and the sections were further stained with hematoxylin. TUNEL-positive cells were counted in five randomly selected fields per slide (×400), and the results were reported as the number of TUNEL-positive cells per 100 cells.

### 4.9. Western Blot Analysis

Prostate tissues underwent homogenization utilizing RIPA lysis buffer (Cell Signaling, Danvers, MA, USA) to facilitate the extraction of proteins. An equal quantity of total protein (15 µg) was subjected to resolution via SDS-PAGE, following which the resolved proteins were transferred onto membranes, which were subsequently incubated overnight with the specified primary antibodies: anti-cyclin D1, anti-cleaved caspase 3, anti-JNK, anti-P-JNK, and anti-β-actin (Cell Signaling). After that, the membranes were exposed to secondary antibodies at room temperature for a duration of 2 h, with the resultant signals being detected through a chemiluminescence detection kit and quantified employing a CSAnalyzer4 (Atto, Tokyo, Japan).

### 4.10. Quantitative Real-Time PCR (RT-qPCR)

Total RNA was isolated from prostate tissues utilizing TRIzol reagent (Thermo Scientific, Waltham, MA, USA). Equivalent quantities of total RNA (1.4 µg) derived from the prostate tissues of each individual rat were exposed to reverse transcription (RT), employing a commercial RT kit (Toyobo, Osaka, Japan). A quantitative real-time PCR was executed utilizing SYBR Green Master Mix (Thermo Scientific, MA, USA) alongside the designated PCR primers. TGF-β, f:5′-AGGGCTCAACACATGGAC-3′ and r: 5′-GGGCCCCACACAGGAGTT-3′; IGF, f:5′-GTCTGTGGCTCAGGCTCTTC-3′ and r: 5′-CCCATTTGGGAACTTCGCCT-3′; FGF, f: 5′-GAGGAAAGGGAAAGGGTCAAAA-3′ and r: 5′-CACAGTGAACGCTCCAGGATT-3′; GAPDH f:5′-CAACTCCCTCAAGATTGTCAGCAA-3′ and r: 5′-GGCATGGACTGTGGTCATGA-3′. The fold change in the expression levels of each target gene in relation to that of the endogenous control (GAPDH) was computed employing the 2^−ΔΔCt^ method.

### 4.11. Statistical Analysis

A series of multiple comparisons were conducted utilizing a one-way analysis of variance (ANOVA), subsequently accompanied by Dunnett’s post hoc test. The threshold for statistical significance was established at *p* < 0.05. The statistical analyses were executed employing GraphPad Prism 9 (GraphPad Software Inc., Boston, MA, USA). The data are represented as the average ± standard deviation (SD).

## 5. Conclusions

The current study demonstrated that the therapeutic treatment of PV extract in rats hindered the development of testosterone-induced BPH, as evidenced by decreases in the weight and hyperplasia of the prostate. This impact can be partly attributed to PV extract-induced decreases in serum levels of DHT and testosterone, as well as its anti-proliferative, pro-apoptotic, and anti-growth properties. These results indicate that PV extract may be utilized as a novel agent for BPH treatment. Further research, such as studies using larger sample sizes, less variable sample knockout models, and/or functional analyses, will be needed to assess the clinical value of PV.

## Figures and Tables

**Figure 1 pharmaceuticals-17-01516-f001:**
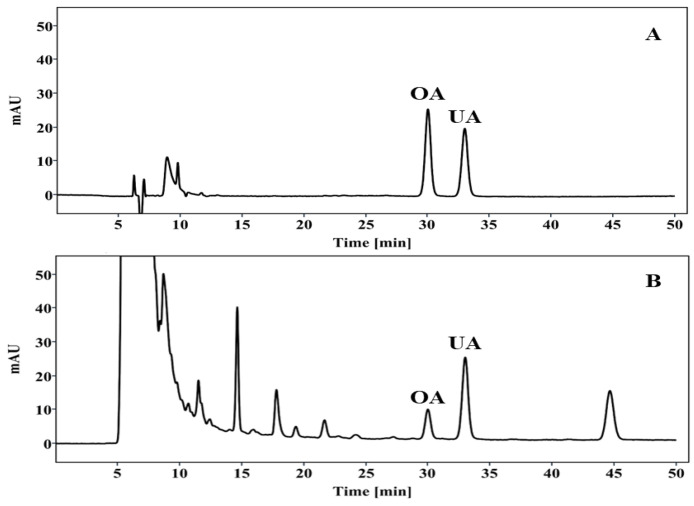
Chromatogram of triterpenoids in PV extract. (**A**) Standard and (**B**) PV extract detected at 210 nm.

**Figure 2 pharmaceuticals-17-01516-f002:**
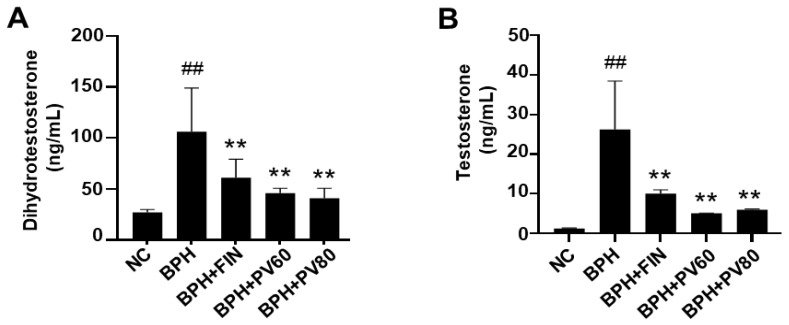
Effects of PV extract on serum levels of dihydrotestosterone and testosterone. (**A**) Serum level of dihydrotestosterone. (**B**) Serum level of testosterone. NC, PBS- and corn oil-dispensed rats; BPH, PBS- and testosterone-dispensed rats; BPH + FIN, testosterone- and finasteride-dispensed rats; BPH + PV60, testosterone- and PV 60 mg/kg-dispensed rats; BPH + PV80, testosterone- and PV 80 mg/kg-dispensed rats. Representation of results as average ± SD. Comparison ## *p* < 0.01 with the NC group; comparison ** *p* < 0.01 with the BPH group.

**Figure 3 pharmaceuticals-17-01516-f003:**
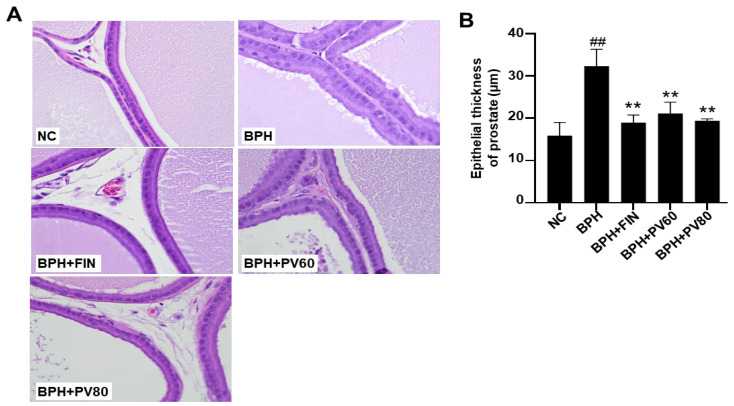
Effects of PV extract on the epithelium thickness of the prostate gland in a rat BPH model. (**A**) Representative images of hematoxylin and eosin (H&E)-stained prostate tissues (magnification, ×400). (**B**) The epithelial thickness of the prostate gland. NC, PBS- and corn oil-dispensed rats; BPH, PBS- and testosterone-dispensed rats; BPH + FIN, testosterone- and finasteride-dispensed rats; BPH + PV60, testosterone- and PV 60 mg/kg-dispensed rats; BPH + PV80, testosterone- and PV 80 mg/kg-dispensed rats. Representation of results as average ± SD. Comparison ## *p* < 0.01 with the NC group; comparison ** *p* < 0.01 with the BPH group.

**Figure 4 pharmaceuticals-17-01516-f004:**
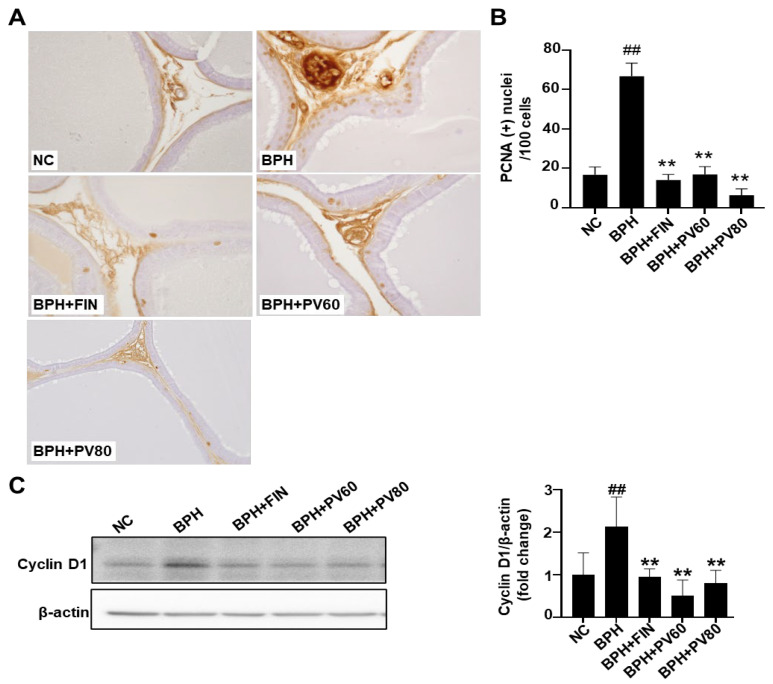
Effects of PV extract on prostate cell proliferation in a rat BPH model. (**A**) Respective images showing immunohistochemical analysis of targeting proliferative cell nuclear antigen (PCNA, magnification, ×400). (**B**) Quantitative analysis of PCNA-positive cells. (**C**) Expression of cyclin D1. NC, PBS- and corn oil-dispensed rats; BPH, PBS- and testosterone-dispensed rats; BPH + FIN, testosterone- and finasteride-dispensed rats; BPH + PV60, testosterone- and PV 60 mg/kg-dispensed rats; BPH + PV80, testosterone- and PV 80 mg/kg-dispensed rats. Representation of results as average ± SD. Comparison ## *p* < 0.01 with the NC group; comparison ** *p* < 0.01 with the BPH group.

**Figure 5 pharmaceuticals-17-01516-f005:**
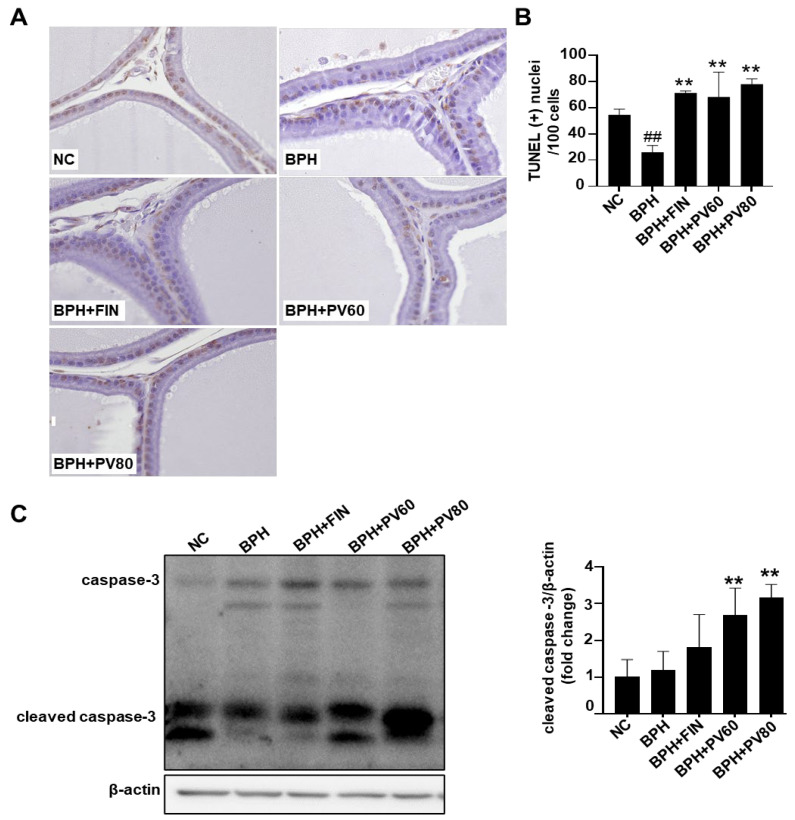
Effects of PV extract on the apoptosis of prostate cells in a rat BPH model. (**A**) Terminal deoxynucleotidyl transferase dUTP nick-end labeling (TUNEL) staining (magnification, ×400). (**B**) As a percentage, the count of TUNEL-positive cells. (**C**) Cleaved caspase-3 expression in prostate tissue. NC, PBS- and corn oil- dispensed rats; BPH, PBS- and testosterone-dispensed rats; BPH + FIN, testosterone- and finasteride-dispensed rats; BPH + PV60, testosterone- and PV 60 mg/kg-dispensed rats; BPH + PV80, testosterone- and PV 80 mg/kg-dispensed rats. Representation of results as average ± SD. Comparison ## *p* < 0.01 with the NC group; comparison ** *p* < 0.01 with the BPH group.

**Figure 6 pharmaceuticals-17-01516-f006:**
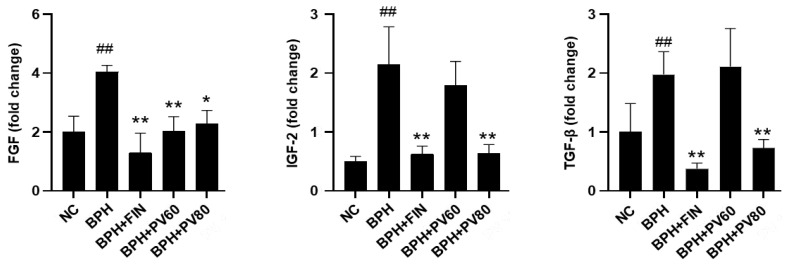
Effects of PV extract on growth factor expression levels in a rat BPH model. The prostatic mRNA levels of FGF, IGF-2, and TGF-β. NC, PBS- and corn oil-dispensed rats; BPH, PBS- and testosterone-dispensed rats; BPH + FIN, testosterone- and finasteride-dispensed rats; BPH + PV60, testosterone- and PV 60 mg/kg-dispensed rats; BPH + PV80, testosterone- and PV 80 mg/kg-dispensed rats. Representation of results as average ± SD. Comparison ## *p* < 0.01 with the NC group; comparison * *p* < 0.05 and ** *p* < 0.01 with the BPH group.

**Figure 7 pharmaceuticals-17-01516-f007:**
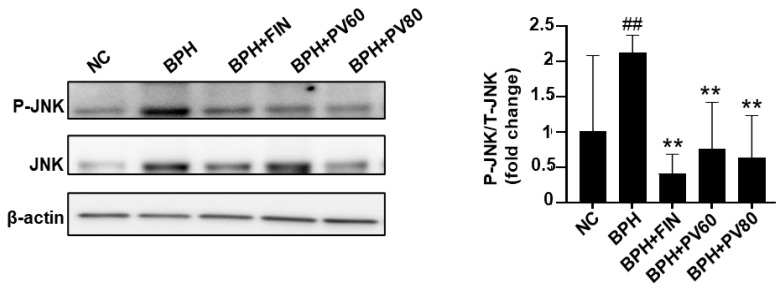
Effects of PV extract on JNK activation in a rat BPH model. Western blot analysis of total JNK (T-JNK) and phosphorylated JNK (P-JNK) expression. NC, PBS- and corn oil-dispensed rats; BPH, PBS- and testosterone-dispensed rats; BPH + FIN, testosterone- and finasteride-dispensed rats; BPH + PV60, testosterone- and PV 60 mg/kg-dispensed rats; BPH + PV80, testosterone- and PV 80 mg/kg-dispensed rats. Representation of results as average ± SD. Comparison ## *p* < 0.01 with the NC group; comparison ** *p* < 0.01 with the BPH group.

**Table 1 pharmaceuticals-17-01516-t001:** Effect of PV extract on prostatic relative weight and inhibition of growth percent.

Group	Treatment	Body Weight (g)	Relative Prostate Weight (%)	Inhibition of Growth (%)
NC	Corn oil + PBS	431.1 ± 19.3	0.185 ± 0.031	-
BPH	TP + PBS	393.2 ± 14.5	0.346 ± 0.075 ^##^	-
BPH + FIN	TP + finasteride	450.3 ± 24.6	0.292 ± 0.059 *	66.49
BPH + PV60	TP + PV 60 mg/kg	406.7 ± 22.2	0.289 ± 0.064 *	35.87
BPH + PV80	TP + PV80 mg/kg	454.1 ± 24.2	0.287 ± 0.053 *	36.64

NC, PBS- and corn oil-dispensed rats; BPH, PBS- and testosterone-dispensed rats; BPH + FIN, testosterone- and finasteride-dispensed rats; BPH + PV60, testosterone- and PV 60 mg/kg-dispensed rats; BPH + PV80, testosterone- and PV 80 mg/kg-dispensed rats. Representation of results as average ± SD. Comparison ## *p* < 0.01 with the NC group; comparison * *p* < 0.05 with the BPH group.

## Data Availability

The data that support the findings of this study are available from the corresponding author upon reasonable request.

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
