# Peer review of "Prunella vulgaris Extract Ameliorates Testosterone-Induced Benign Prostatic Hyperplasia by Regulating Androgen Levels, Cell Proliferation, and Apoptosis"

_pharmaceuticals, 2024, doi:10.3390/ph17111516_

Round 1

Reviewer 1 Report

Comments and Suggestions for Authors

Prunella Vulgaris Extract Ameliorates Testosterone-Induced Benign Prostatic

Th overall goal of this study was to assess the effectiveness of Prunella vulgaris L. (PV), a perennial herbaceous plant native to Europe and Asia in the growth of Benign prostatic hyperplasia (BPH). The study now concluded that PV effectively ameliorated the development of testosterone-induced BPH.

This study is interesting but it’s not something new. Previous research has shown that PV has anti-cancer properties so it is unclear as to what is the innovation in this study.

Although the last paragraph in the introduction stated that no studies have examined the potential of PV to prevent BPH, previous manuscripts have shown its effectiveness in tumor including the following below:

1.      Huang, M., Wang, Y., Xu, L. et al. Anti-tumor Properties of Prunella vulgaris . Curr Pharmacol Rep 1, 401–419 (2015). https://doi.org/10.1007/s40495-015-0038-6

2.      Prunella vulgaris: A Comprehensive Review of Chemical Constituents, Pharmacological Effects and Clinical Applications https://www.ingentaconnect.com/content/ben/cpd/2019/00000025/00000003/art00014#Refs

3.      Ya-gang Song, Le Kang, Shuo Tian, Lin-lin Cui, Yan Li, Ming Bai, Xiao-yan Fang, Li-hua Cao, Kimberly Coleman, Ming-san Miao, Study on the anti-hepatocarcinoma effect and molecular mechanism of Prunella vulgaris total flavonoids, Journal of Ethnopharmacology,

Volume 273, 2021,113891, ISSN 0378-8741, https://doi.org/10.1016/j.jep.2021.113891.

I think it is best to at least do some additional background information on what was done in previous studies related to BPH and its effectiveness in treating cancer in some studies.

In some figures, please make sure to define the acronyms before using these.

Overall, I think this study was clearly written and organized. The figures are very clear.

It is unclear as to how many rats were used in the study.

Reviewer 2 Report

Comments and Suggestions for Authors

The article by Kumbukgahadeniya et al elucidates the positive effects of Prunella Vulgaris (PV) extract on the model Benign Prostatic Hyperplasia (BPH) induced in mice by testosterone injection. The authors obtained interesting results. HPLC analysis revealed oleanolic (OA) and ursolic (UA) acids as the main components of PV-extracts.  PV-extract treatment significantly normalized the steady-state of model BPH mice: the prostate weight, serum levels of testosterone and DHT, nuclear antigen (PCNA) expression and cyclin D1 expression, m-RNA expression levels of growth factors FGF, IGF-2, TGF-β were also significantly attenuated as was shown by the authors with great statistical significance (p<0.01).

At the same time there are shortcomings, the elimination of which would significantly improve the content of the article.

 1. Materials and Methods must contain information about preparation of PV extract and explanation about doses 60mg and 80mg (of dried extract?)  and solvent used. 60mg/kg dose, for example, was effective to reduce expression of only FGF growth factor while growth factors IGF-2, TGF-β were significantly stimulated. In many other cases the dose 80mg/kg was more effective compared with 60mg/kg. So why not to use higher doses – the point should be at least discussed. Also should be elucidated side effects of PV-extract if such are known to the authors.

2.  Positive effects of PV-extract were compared with effects of finasteride, a standard medicine,  used to treat BPH, which has a number of adverse effects. These effects should be listed and it would be logic to show if PV-extract has or does not have the same.

3.  As shown on Fig.5C PV extract effectively increase the level of cleaved caspase-3 expression in prostate cells in BPH model mice. According to representative columns (right part of Fig5C) the column NC (control) does not differ much from column BPH while after treatment with finasteride or PV-extract the columns significantly increased. Is the effect positive or results in disbalans between proliferation and apoptose – the point should be discussed.

The article is physiologically relevant and should be published in Pharmaceuticals after small revision.  

Reviewer 3 Report

Comments and Suggestions for Authors

This study investigates the therapeutic potential of Prunella vulgaris L. extract against Benign prostatic hyperplasia development in a testosterone-induced BPH rat model. Overall, the manuscript looks good. However, it can be enhanced in certain areas.

1.     I suggest the authors expand the information supplied in lines 40-42. Discuss the role of 5α-reductases in converting testosterone into DTH including how the affinity of the latter to androgen receptor mediates the BPH development. This information supports the information mentioned in lines 47-51.

2.     Are there any side effects of treatment options presented in lines 47-51? I suggest mentioning the adverse effects. These effects could be the answer to why the authors are interested in plant extract-based therapeutics.

3.     Since the authors purchased PV from the herbal market, how did they identify the PV?

4.     How did the authors acquire the standards mentioned in line 282? What was the purity of these standards? And, what is the relative content of OA and UA in the extracts of PV compared to these standards?

5.     Had the authors administrated a single batch of the PV extract for 4 weeks? It is not clear from the methods mentioned in sub-section 4.4.

6.     Mention if there is any significant difference in the body weight of rats among the four groups in sub-section 2.2.

7.     Had the authors studied the sperm counts of rats among the groups? It could be useful to compare the effect of FIN with PV.

8.     I recommend that the authors correlate their findings with the existing literature. How do their results differ from other kinds of plant extracts or clinical drugs?
